# FundusAdapter: few-shot adaptation of fundus image foundation model for fundus image diagnosis

**Yifan Chang**[1,2]                                    YIFAN_CHANG@MAIL.USTC.EDU.CN

**Zihang Jiang**[1,2]                                    JZH0103@USTC.EDU.CN

**Kun Zhang**[*1,2]                                    KKZHANG@USTC.EDU.CN

**S Kevin Zhou**[*1,2,3,4]                                    S.KEVIN.ZHOU@GMAIL.COM

[1] *School of Biomedical Engineering, Division of Life Sciences and Medicine, University of Science and Technology of China (USTC), Hefei Anhui, 230026, China*

[2] *Center for Medical Imaging, Robotics, Analytic Computing & Learning (MIRACLE), Suzhou Institute for Advance Research, USTC, Suzhou Jiangsu, 215123, China*

[3] *Jiangsu Provincial Key Laboratory of Multimodal Digital Twin Technology, Suzhou Jiangsu, 215123, China*

[4] *State Key Laboratory of Precision and Intelligent Chemistry, USTC, Hefei Anhui 230026, China*

## Abstract

Fundus images exhibit significant source gaps, limiting the performance of foundation models across different scenarios. Due to the scarcity of labeled training data, few-shot adaptation is essential for effective diagnosis. However, existing few-shot adapters have primarily focused on global image features, which are insufficient for distinguishing fundus diseases that require detailed texture information. In this paper, we propose FundusAdapter, the first few-shot adaptation model of fundus image foundation model for fundus image diagnosis. By leveraging hierarchical feature extraction, FundusAdapter effectively integrates both global and local features, enhancing the detection of subtle lesions. The use of cross-attention and gate memory guidance improves the interaction between features, leading to more accurate adaptation. Our model achieves state-of-the-art performance on public fundus benchmarks. Code is available at https://github.com/Yifan-Chang/CrossFundus.

**Keywords:** Few-shot adaption, Fundus image, Foundation model.

## 1. Introduction

Fundus imaging is one of the most common and crucial diagnostic tools, not only reflecting the condition of the retina and eye but also serving as an indicator of systemic diseases such as diabetes (Zhou et al., 2023). Given the diversity of fundus-related conditions, there has been an increasing attention on image-text alignment model (Zhang et al., 2023, 2022a), especially foundation models, which offer flexible classification scalability and zero-shot generalizability, in fundus research. While source differences exist due to factors such as device variability and race, and the scarcity of medical images (Moris et al., 2024), adapting models to data distributions that differ from the training data in few-shot scenarios remains an issue that warrants further exploration. Shakeri et al. (Shakeri et al., 2024) show the performance of the existing method(clip adapter(Gao et al., 2024), LP++(Huang et al., 2024) etc.) on fundus image few-shot cross-source adaption. Generally, all of these approaches leverage global image features to align with text or cached features. However,

---

[*] Corresponding authors

it is not reasonable to rely solely on global features in the fundus few-shot adaptation task. Several studies (Shakeri et al., 2024; Silva-Rodriguez et al., 2025) highlight the limitations of existing VLM adaptation models in the context of few-shot adaptation for fundus images, using the global-level features, proved with some even performing worse than a simple linear probe. Moreover, both local and global features are essential for the diagnosis of fundus images. For example, hard exudates and cotton wool spots are key indicators of diabetic retinopathy, whereas cataracts require global features for diagnosis. To address this issue, we propose a novel framework for few-shot adaptation of pretrained fundus image foundation model, named FundusAdapter for fundus image diagnosis.

## 2. Method

Figure 1 shows the framework of our method. Firstly, the fundus image, disease description, and local image patches are encoded by pre-trained VLM FLAIR(Silva-Rodriguez et al., 2025). Cross attention module gives a primary metric depending on both local and global features. Then gate memory will help hierarchical cross guidance for further metric learning. Finally, prediction will be given by the linear combination of different modules.

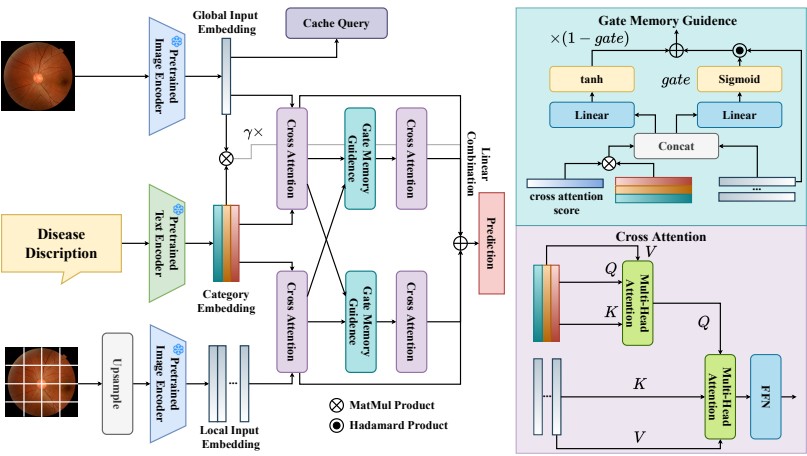

Figure 1: The overall framework of FundusAdapter.

Given a fundus image input $X$ and the text for each category $Y$, the features of the global image, local patches, and query text categories are denoted as:$I^{Global} = \Phi_{image}(X)$; $I^{Global} \in \mathbb{R}^{1 \times d}$;$I^{Local} = \text{Concat}(\Phi_{image}(\text{Upsample}(X_i)))$; $I^{Local} \in \mathbb{R}^{m \times d}$;$T = \text{Concat}(\Phi_{text}(Y_j))$; $T \in \mathbb{R}^{c \times d}$. Where $X_i$ and $Y_j$ represent each image patch and the text description for each category (from FLAIR(Silva-Rodriguez et al., 2025)), respectively. $d$ denotes the embedding dimension of the encoder, $m$ refers to the number of local patches in a single image, and $c$ represents the number of query categories (including both normal and diseased categories). We use a parallel structure to process global input and local input respectively and obtain $P_{init}^{Global}$ and $P_{init}^{Local}$ by the first cross attention module. Gate memory guidance module use $P_{init}^{Global}$ and $P_{init}^{Local}$ to interact with global and local information and update the representation of the image. $P_{new}^{Global}$ and $P_{new}^{Local}$ denote the outputs of the cross-attention module with $I^{Local}$ and $I^{Global}$ replaced by $I_{new}^{Local}$ and $I_{new}^{Global}$,

respectively. The final prediction is the linear combination of outputs of different modules as: $pre = \gamma \frac{I^{Global}T^\top}{||I^{Global}||\cdot||T||} + P_{cache} + (1 - \frac{1}{m+1})P^{Global} + \frac{1}{m+1}P^{Local}$, where $P^{Global} = (1 - \frac{1}{m})P_{init}^{Global} + \frac{1}{m}P_{new}^{Global}$ and $P^{Local} = (1 - \frac{1}{m})P_{init}^{Local} + \frac{1}{m}P_{new}^{Local}$. A cache query module $P_{cache}$ is designed referring to Tip-Adapter (Zhang et al., 2022b). The coefficients in front of cross-attention are scale factors designed to balance the effects of global and local features. $\gamma$ is a learnable parameter. The cross-entropy loss function is used to optimize the classification model: $Loss = -\sum_j^c y_j \log(pre_j)$. $y_j = 1$ if $j$ equals to the ground-truth category label, otherwise $y_j = 0$. $pre_j$ is predicted probability for class $j$.

## 3. Results and Discussion

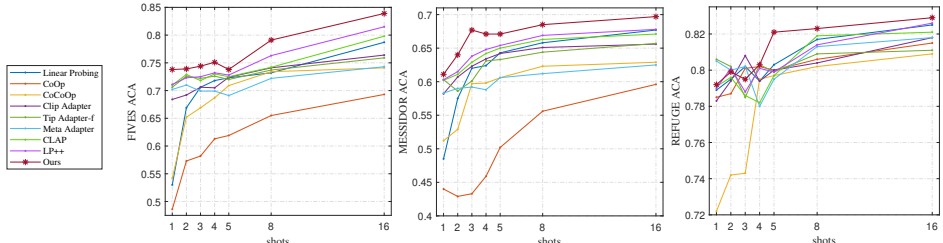

Figure 2: Comparison of our model and other methods on benchmark

Using FLAIR as our image and text encoder, we have to carefully select datasets that are out of training dataset of FLAIR. Three common used public fundus datasets: FIVES(Jin et al., 2022) , MESSIDOR(Decencière et al., 2014), and REFUGE(Orlando et al., 2020) are chosen by us to evaluate our method. Previous few-shot adapter models(Zhou et al., 2022b,a; Gao et al., 2024; Zhang et al., 2022b; Song et al., 2023; Huang et al., 2024; Silva-Rodríguez et al., 2024) are compared with our models. We evaluate all the models in a equal environment. Following the experimental setup in FLAIR, metrics are averaged across 5 cross-validation folds. Balanced average accuracy (ACA) is utilized by us as model metrics. The number of local patches $m = 4$, which means we divide the image into four equal parts. FundusAdapter obtain best performance in FIVES and MESSIDOR. In REFUGE, it also do best on 4-16 shots. We also study influence of $m$ on different shots adaption in Table 1. As the number of shots increases, the selection of the optimal $m$ value decreases, which means that the fewer shots there are, the finer-grained local patches are required.

Table 1: Influence of $m$ on ACA of different shots adaption. M: MESSIDOR; F: FIVES

| $m$ | 4 | 9 | 16 | 25 | $m$ | 4 | 9 | 16 | 25 |
|---|---|---|---|---|---|---|---|---|---|
| M-1-shot | 0.608 | **0.631** | 0.629 | 0.622 | F-1-shot | 0.738 | 0.732 | 0.737 | **0.742** |
| M-3-shot | **0.677** | 0.668 | 0.660 | 0.669 | F-3-shot | 0.744 | **0.756** | 0.731 | 0.737 |
| M-5-shot | **0.681** | 0.666 | 0.669 | 0.670 | F-5-shot | **0.738** | 0.735 | 0.731 | 0.730 |

## Acknowledge

This work is supported by the National Natural Science Foun-dation of China under Grant 62271465,the China Postdoctoral Science Foundation under Grant 2024M763178, and the-Suzhou Basic Research Program under Grant SYG202338.

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
