# OpenReview forum: "FundusAdapter: few-shot adaptation of fundus image foundation model for fundus image diagnosis"
_MIDL.io/2025/Short_Papers — MIDL 2025 - Short Papers_

### Official Review · Reviewer_nkoq · 2025-04-25

**Rating:** 5
**Confidence:** 5

**Summary:**

FundusAdapter is a few-shot adaptation framework designed for fundus image diagnosis. It combines global and local features using hierarchical cross-attention and gate memory guidance to better capture subtle disease indicators. The model outperforms previous few-shot adapters on public fundus benchmarks like FIVES, MESSIDOR, and REFUGE.

**Strengths:**

A key strength of FundusAdapter is its innovative use of both local and global features, which is crucial for fundus disease detection that requires fine texture details. Its hierarchical design and strong cross-attention mechanism lead to significant improvements over existing few-shot adaptation methods.

**Weaknesses:**

A limitation of the method is its reliance on carefully tuned patch division (the number of local patches "m"), which adds complexity to adapting the model across different datasets. Additionally, performance can vary depending on the choice of shots and patch granularity.

---

### Decision · Program_Chairs · 2025-05-01

Accept